# Burst firing creates an attractor in synaptic weight dynamics

**Kathleen Jacquerie**[1,2]*, **Danil Tyulmankov**[3,4], **Pierre Sacré**[2], **Guillaume Drion**[2]

**1** Biology Department, Brandeis University, Waltham, Massachusetts, United States of America,
**2** Department of Electrical Engineering and Computer Science, University of Liège, Liège, Belgium,
**3** Viterbi School of Engineering, University of Southern California, Los Angeles, California, United States of America, **4** Center for Theoretical Neuroscience, Columbia University, New York, New York, United States of America

* kathleen.jacquerie@gmail.com

## Abstract

Neural circuits often alternate between tonic and burst firing, two distinct activity regimes that reflect changes in excitability and neuromodulatory state. While tonic firing produces asynchronous spikes driven by diverse external inputs, collective burst firing consists of rapid clusters of spikes followed by a period of silence, happening synchronously within the network. Synaptic plasticity has typically been studied only in either one of these regimes, leaving unclear how their distinct plasticity dynamics can be combined when circuits alternate between regimes. Here, we use a conductance-based network model endowed with calcium-based or spike-timing–based plasticity rules to examine how synaptic weights evolve across tonic and burst firing regimes. During tonic firing, synaptic weights are driven by the statistics of external inputs, producing a broad distribution across the network. In contrast, during collective burst firing, weights converge to a narrow region in weight space: a burst-induced attractor. We derive the location of this attractor analytically in terms of plasticity parameters and activity statistics, and confirm its emergence across diverse plasticity rules. The attractor reflects the synchronization of plasticity-driving signals during bursts, which homogenizes synaptic dynamics and forces convergence toward shared fixed points. We further show that neuromodulation and synaptic tagging can shift or split the burst-induced attractor, stabilizing selected synapses while weakening others. Together, these results identify burst-induced attractors as a robust emergent property of collective bursting. Alternation between tonic and burst firing provides a biologically plausible context in which heterogeneous, input-driven synaptic configurations formed during tonic activity can be selectively consolidated or down-selected by the burst-induced attractor during subsequent bursts. By showing how they can be analytically predicted and experimentally modulated, our work provides a general computational framework linking firing state transitions, synaptic plasticity, and memory organization.

**Data availability statement:** The code files are freely available at https://github.com/KJacquerie/Burst-Attractor.

**Funding:** This work was supported by the Fonds de la Recherche Scientifique—FNRS (grant ASP40006590 to KJ) and by the Belgian Government through the Federal Public Service Policy and Support, under grant NEMODEI2 (support to PS, GD). KJ received a salary from the Fonds de la Recherche Scientifique—FNRS (grant ASP40006590). The funders had no role in study design, data collection and analysis, decision to publish, or preparation of the manuscript.

**Competing interests:** The authors have declared that no competing interests exist.

## Author summary

Brains operate in different activity states, reflecting different behaviors or neuromodulatory states. Neurons can fire isolated spikes in a tonic mode that encodes information about external inputs. They can fire rapid bursts of spikes, generating large synchronized oscillations that dominate population activity. Both tonic and burst firing are linked to learning and memory, yet their distinct contributions to shaping synaptic plasticity remain poorly understood. In this study, we use biophysical network models equipped with well-established plasticity rules to investigate how synaptic weights evolve under tonic and burst firing. We show that during tonic activity, synapses diverge toward a wide variety of values, reflecting the diversity of input statistics. In contrast, when the network enters a collective bursting state, synaptic weights collapse into a narrow region of weight space, a "burst-induced attractor." We derive the attractor mathematically and show that its position depends directly on the plasticity parameters, meaning it can be shifted or split through neuromodulatory and tag-dependent processes. Our results suggest that burst firing defines a robust and controllable consolidation stage, and that alternation between tonic and burst firing provides a natural biological context for linking distinct activity regimes to learning and memory processes.

## Introduction

Neural circuits can alternate between distinct neuronal firing regimes—most notably either tonic or burst firing—depending on behavioral demands and neuromodulatory state [1]. In tonic firing, neurons emit relatively isolated spikes that are primarily driven by ongoing external inputs. At the population level, this activity corresponds to low-amplitude, high-frequency oscillations in local field potential (LFP) or electroencephalogram (EEG) recordings. In contrast, burst firing consists of rapid clusters of spikes followed by periods of silence. Bursts can emerge endogenously from intrinsic neuronal and network properties, reflecting cycles of ion channel activation and inactivation, like T-type calcium channels [2–6]. When many neurons burst synchronously, the resulting collective dynamics generate large-amplitude, low-frequency LFP/EEG signals characteristic of highly synchronized activity. This dynamic switching between tonic and burst regimes is observed across multiple neural systems and is thought to support different functional states of the brain. Tonic firing is typically associated with active sensory processing and learning [7,8], whereas burst firing is more often observed during quiet behavioral states that facilitate memory consolidation [1,9–12].

Neural circuits store information via synaptic plasticity—activity-dependent changes in synaptic strength. These changes involve multiple mechanisms, including increased postsynaptic receptor efficacy, the insertion of additional receptors via exocytosis, *de novo* protein synthesis, and structural remodeling of synapses [13–16]. Synaptic plasticity, however, has largely been studied under either tonic or burst

firing in isolation, and it remains unclear whether these regimes drive different forms of information encoding or learning; whether tonic and burst firing support distinct modes of synaptic weight change, or whether they engage overlapping mechanisms. Our central question is therefore: how do synaptic weights evolve between tonic and burst firing regimes, and what implications does this have for learning and memory?

This question has largely been unexplored in computational models, primarily for two reasons. First, it requires a network model capable of robustly alternating between firing modes despite variability in neuronal and synaptic properties. Second, it requires a plasticity rule that remains valid across distinct activity regimes. Most existing models address plasticity in only one regime at a time [17]. Many rely on spike-timing–dependent plasticity (STDP) formulations -- pair-based or triplet rules [18–20] -- or on calcium-based plasticity models [21–23], which are typically tested under tonic firing conditions. Burst-related plasticity has also been explored using burst-timing–dependent rules [24]. However, neither approach has been examined in the context of networks switching between firing states, despite the prevalence of such state transitions in the brain.

To address this gap, we build on our previous work, which introduced a biophysical, conductance-based network that robustly switches between tonic and burst firing [5] allowing us to examine general principles of synaptic plasticity across activity regimes. Here we endow that network with synaptic plasticity using a calcium-based rule [22,25] and examine, in parallel, representative spike-based rules (pair-based and triplet STDP).

We find that when the network enters a collective bursting state, synaptic weights converge to a narrow region in weight space—a burst-induced attractor. The same qualitative phenomenon emerges across calcium- and STDP-inspired rules, and we derive the attractor location analytically. Crucially, the attractor follows directly from the plasticity parameters, so it can be modified by neuromodulatory and tag-dependent processes. This turns the attractor into a feature: during bursting, adjusting plasticity parameters globally or in a tag-dependent manner can shift or split the attractor, enabling selective consolidation (e.g., stabilizing tagged synapses while down-selecting untagged ones). More broadly, this frames bursting as a plausible controllable consolidation stage.

Overall, our results argue that plasticity should be studied in networks that operate across tonic and burst firing regimes. Together, these regimes provide a general mechanism by which collective burst firing organizes synaptic weights and supports consolidation.

## Results

### Modeling robust transitions between tonic and burst firing in a biophysical network model

We simulate the transition from tonic to burst firing in a heterogeneous network of conductance-based model neurons (Materials and methods), with the network architecture shown in Fig 1A. The network is composed of $N$ presynaptic excitatory neurons projecting to $M$ postsynaptic excitatory neurons with membrane potentials $V_j$ and $V_i$, respectively. These neurons all receive GABA currents from a single inhibitory neuron ($V_{inh}$), the activity of which controls whether the network is in a tonic firing or bursting regime. This modulation in inhibitory levels mimics the variation of neuromodulators such as acetylcholine, dopamine, serotonin, or histamine [26], known to control brain state transitions [5,27–31]. Each neuron also receives an individual applied current ($I_{app,inh}$, $I_{app,j}$, $I_{app,i}$) to set its firing rate for learning specified input patterns. For details on equations governing neural dynamics, see Materials and methods.

Fig 1B illustrates the transition between the two firing regimes in a network of $50$ presynaptic and $50$ postsynaptic neurons: *input-driven tonic firing* and *collective burst firing*, controlled by the current $I_{app,inh}$ applied to the inhibitory neuron.

During *input-driven tonic firing*, a depolarizing current $I_{app,inh}$ applied to the inhibitory neuron induces regular tonic spiking activity, as commonly observed in inhibitory pacemaking cortical neurons [1]. Consequently, this pacemaking inhibitory neuron sets the resting membrane voltage of all the excitatory neurons. Upon receiving an applied current ($I_{app,j}$, $I_{app,i}$), these excitatory neurons generate action potentials. For details on protocol inducing tonic firing, see Materials and methods.

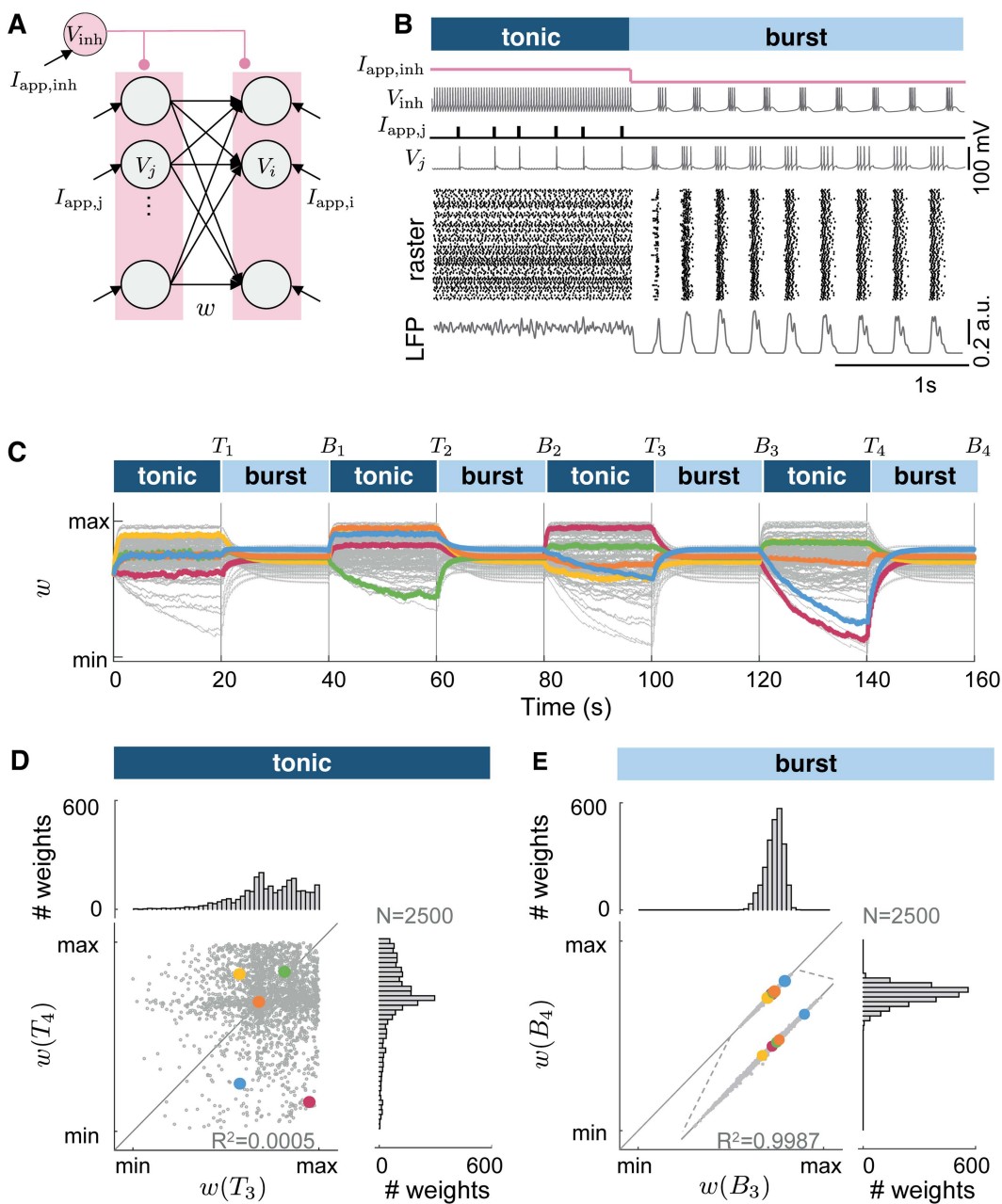

**Fig 1. Collective burst firing drives synaptic weights towards attractor in the weight space. A.** The network model consists of one inhibitory neuron ($V_{inh}$) projecting to all the excitatory neurons, connected in a feedforward configuration with $N$ presynaptic neurons ($V_j$) to $M$ postsynaptic neurons ($V_i$). An external current applied to the inhibitory neuron controls the network state ($I_{app,inh}$). Each excitatory neuron is subjected to an individual external current mimicking an external input. **B.** Hyperpolarizing the current applied to the inhibitory neuron ($I_{app,inh}$) switches the network state from input-driving tonic firing to collective burst firing. Raster plot activity of the excitatory neurons and local field potential (LFP) traces are displayed. **C.** Evolution of the synaptic weights during switches from input-driven tonic firing to collective burst firing. The network comprises 50 presynaptic neurons connected to 50 postsynaptic neurons for 100 traces among 2500 (gray lines) with 5 randomly highlighted lines (colored lines). $T_k$ and $B_k$ indicate the end of the $k$-th tonic and burst firing state, respectively. **D-E.** Comparison of the synaptic weights at the end of the third and fourth tonic firing states (dark blue) or the third and fourth burst firing states (light blue), normalized between the minimal and maximal values shown on a scatter plot. The histograms highlight the weight distribution. The weights converge to a fixed point in the weight space, we call burst-induced attractor ($\min(w_{ij}) = 0.176$, $\max(w_{ij}) = 0.705$).

To trigger a network transition from input-driven tonic firing to *collective burst firing*, a hyperpolarizing current $I_{app,inh}$ is applied to the inhibitory neuron.

This bursting activity, in which each neuron generates a rapid succession of action potentials followed by a quiescent period [32–34], is generated endogenously by intrinsic ion channel dynamics and recurrent inhibition: once in this regime, we silence external inputs in $I_{app,j}$ and $I_{app,i}$ and retain only a small noisy baseline current to the excitatory neurons, which does not by itself drive spiking. Thus, the timing and occurrence of bursts are not imposed by external patterned input, but emerge from the intrinsic properties of the neurons, such as T-type calcium channels that deinactivate when the neuron is hyperpolarized. Sufficiently strong or temporally structured external inputs could modulate the burst pattern, but the burst regime characterized here does not require such inputs. The term "collective" refers to the synchronization of individual bursting neurons despite heterogeneity in intrinsic properties and synaptic connections, resulting in slow large-amplitude population activity shown in the LFP oscillation, contrasting the fast small-amplitude observed in input-driven tonic firing (Fig 1B, LFP traces) [5]. For details on LFP calculation, see Materials and methods.

### Modeling synaptic plasticity

Synapses between presynaptic and postsynaptic excitatory neurons undergo activity-dependent plasticity, a fundamental process underlying learning and memory. In biological systems, plasticity arises from a combination of biochemical, structural, and electrical mechanisms that respond to intracellular calcium dynamics.

To capture these dynamics quantitatively, we use a state-of-the-art calcium-based model established by Graupner et al. [25], which was fit to experimental data obtained through a frequency pairing protocol [35]. The model implements two opposing calcium-triggered pathways that lead to either an increase or a decrease in synaptic strength: potentiation or depression is triggered when calcium levels exceed specific thresholds. Although we illustrate our results with this model, we later show that the same principles apply across a wide range of plasticity rules, highlighting the generality of our results. In this framework, the synaptic weight from the presynaptic neuron $j$ to the postsynaptic neuron $i$ is denoted $w_{ij}$ and its evolution is governed by Eq (3). For details on equations governing synaptic plasticity, see Materials and methods.

### Collective burst firing drives synaptic weights towards an attractor in the weight space

To study how synaptic plasticity operates across tonic and burst firing regimes, we track the evolution of the weights over the course of eight successive tonic and burst epochs (Fig 1C).

During tonic firing (Fig 1C, dark blue), each neuron receives a random pulse train input, with the stimulation frequency selected uniformly at random between 0.1 and 50 Hz. For details on the computational experiment, see Section Computational experiment related to Fig 1. The temporal evolution of the synaptic weight $w_{ij}(t)$ is determined by the correlation between the activity of presynaptic neuron $j$ and postsynaptic neuron $i$, which drives plasticity at those synapses. As a result, at the end of each tonic firing period (time $T_k$, $k = 1, \ldots, 4$), the weight $w_{ij}(T_k)$ converges to a value determined by the specific external input received. Since different neurons receive different input statistics, the weights across the network settle at a broad range of values (see Fig 1D, histograms). Furthermore, because the input statistics vary across tonic periods, the synaptic weights also differ from one tonic episode to the next, as illustrated by plotting $w(T_4)$ against $w(T_3)$ (Fig 1D, scatter plot; see also S1 Fig for comparison across other states).

During burst firing (Fig 1C, light blue), the network is driven only by an inhibitory current, with no external input to the excitatory neurons. Collective bursting activity drastically alters synaptic weight dynamics: previously potentiated connections tend to depress, and vice versa. Each initial weight $w_{ij}(T_k)$ converges to a unique steady-state value $w_{ij}(B_k)$ within a narrow range (Fig 1E, histograms), and repeats from one burst episode to the next one independent of its initial condition, as illustrated by plotting $w(B_4)$ against $w(B_3)$ (Fig 1E, scatter plot; see S1 Fig for other pairs). We call this phenomenon a *burst-induced attractor* and define it as a narrow region in weight space to which synaptic weights converge under collective bursting dynamics.

## The burst-induced attractor emerges in a wide range of synaptic plasticity models

We next investigate whether the choice of synaptic plasticity rule influences the emergence of a burst-induced attractor. We repeat the same experiment in the same network using various synaptic plasticity models and observe that the effect persists across a wide range of rules (S2 Fig), including different variants of calcium-based models [21–23], phenomenological models based on pairwise [18] or triplet [20] spike timing, weight-dependent rules [36,37], and models with parameters fitted to cortical frequency-based protocols [35] or hippocampal spike-based protocols [38]. We also compare the influence of weight dependence by testing both soft and hard bounds in the synaptic rule. During collective burst firing, synaptic weights converge towards an attractor in the weight space no matter the choice of the synaptic plasticity rule.

## Analytical insight into the burst-induced attractor

Why does collective burst firing constrain synaptic weights into an attractor, whereas tonic firing produces a broad spread? To address this question, we derive the fixed points of synaptic weight dynamics under calcium-based and spike-timing–based plasticity rules for a given firing regime and then verify these predictions in network simulations.

We first focus on calcium-based plasticity rules, and we derive the fixed point for one of the rules, [22]. Two thresholds determine the outcome of calcium dynamics: a depression threshold $\theta_d$ and a potentiation threshold $\theta_p$. Each calcium regime is associated with a steady-state value (potentiation or depression) and a characteristic relaxation timescale. When calcium remains below $\theta_d$ ($c_{ij} < \theta_d$), the weight does not change. For intermediate calcium levels ($\theta_d \leq c_{ij} < \theta_p$), the weight drifts toward the depression steady state $\Omega_d$ with time constant $\tau_d$. When calcium exceeds $\theta_p$ ($c_{ij} \geq \theta_p$), the weight relaxes toward the potentiation steady state $\Omega_p$ with time constant $\tau_p$. As shown in Fig 2A, calcium fluctuations can therefore be decomposed into potentiation, depression, and neutral intervals. By averaging the fast calcium dynamics, we obtain the fixed point of the synaptic weight:

$$\bar{w}_{ij} = \frac{\Omega_d \alpha_{d,ij} + \Omega_p \alpha_{p,ij}}{\alpha_{d,ij} + \alpha_{p,ij}}.$$

(1)

Here, $\alpha_{d,ij}$ and $\alpha_{p,ij}$ denote the effective times spent in the depression and potentiation regimes, respectively. Eq (1) is valid for synapses that spend non-zero time in the depression or potentiation regimes, i.e., when $\alpha_{d,ij} + \alpha_{p,ij} > 0$. When there is neither depression nor potentiation ($\alpha_{d,ij} = \alpha_{p,ij} = 0$), synaptic plasticity is inactive and Eq (1) does not apply; in that case the synaptic weight remains unchanged and $\bar{w}_{ij} = w_{ij}(0)$, where $w_{ij}(0)$ is the weight at the onset of the firing state (or equivalently at the switch between firing states). This derivation is based on previous works [37,39]. For more details on the derivation, see S1 Text D.

A similar principle holds for STDP rule, where correlations between pre- and postsynaptic spikes reflect how much time is spent in potentiation versus depression window. For the STDP model of [19], the fixed point is:

$$\bar{w}_{ij} = \frac{\dfrac{A^+ C^+}{A^- C^-}}{1 + \dfrac{A^+ C^+}{A^- C^-}},$$

(2)

where $A^+$ and $A^-$ are potentiation and depression parameters of the STDP rule, and $C^+$ and $C^-$ are measures of the synaptic potentiation and depression that stems from all input–output correlations with positive and negative time lag, respectively. For more details on the derivation, see S1 Text E. The results for hard-bounds can be found in S1 Text F.

In both cases, the fixed point depends on the plasticity parameters and on activity statistics, time spent in calcium regimes or spike correlations [37,40].

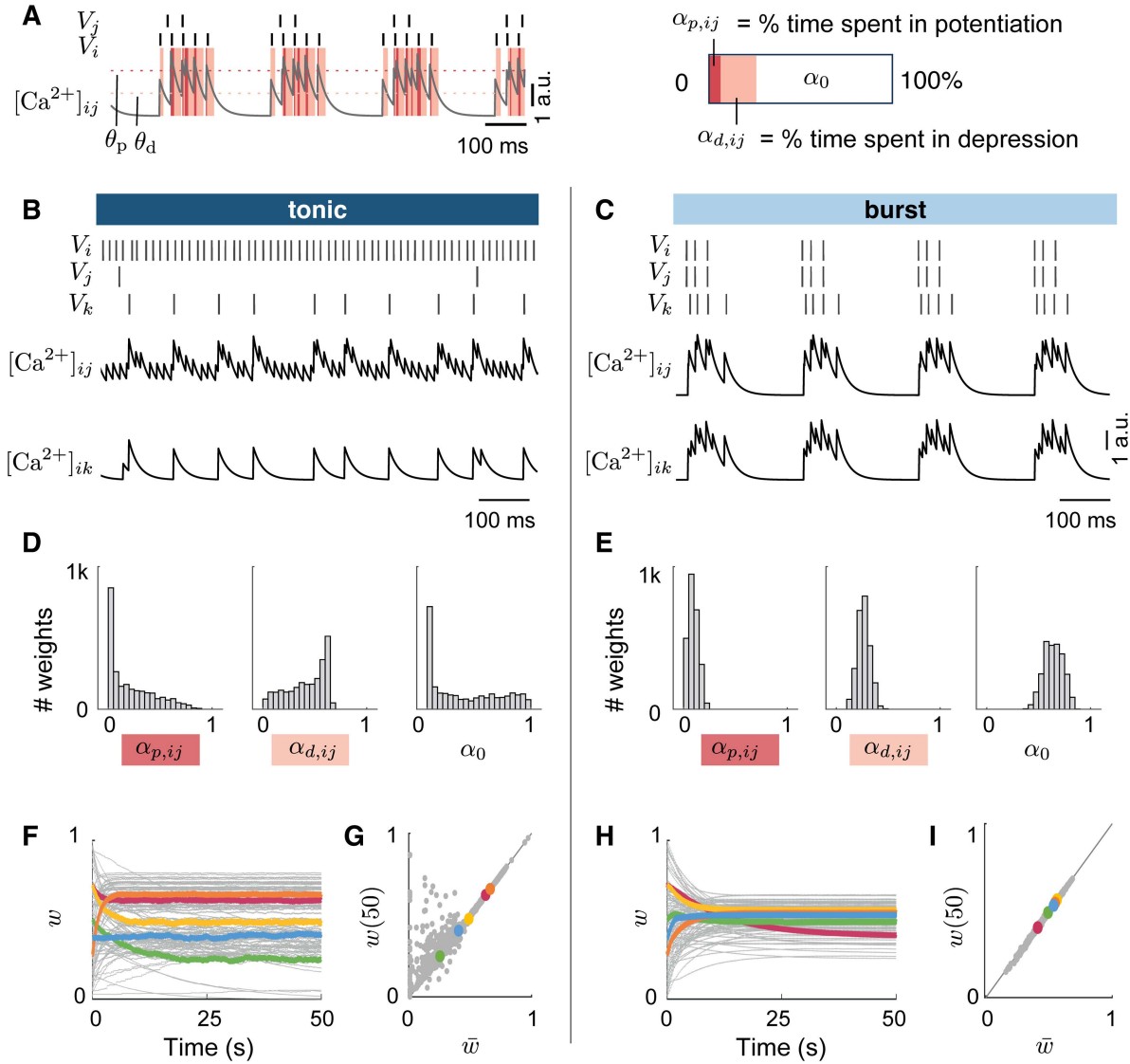

**Fig 2. Burst firing constrains synaptic weights toward an attractor via shared calcium dynamics. A.** Calcium trace decomposition into potentiation, depression, and neutral zones, based on pre- and postsynaptic spike timing. Red and orange shading indicate time spent above the potentiation threshold or below the depression threshold, respectively. Right: effective time in each region, weighted by corresponding time constants ($\alpha_{p,ij}$, $\alpha_{d,ij}$). **B-C.** Example spike trains and calcium traces for two synapses during tonic and burst firing. Tonic activity produces diverse calcium dynamics, while burst firing induces synchronized fluctuations across synapses. **D-E.** Distribution of effective time allocated in different plasticity regions during tonic and burst firing. Tonic firing shows broad distributions, while burst firing produces narrower, more consistent distributions across synapses. **F-H.** Evolution of synaptic weights over time during tonic (F) and burst (H) firing, starting from the same initial conditions. Weights diverge toward distinct values under tonic input, whereas they converge to a narrow range under burst firing, indicating the emergence of a burst-induced attractor. **G-I.** Predicted steady-state weight using analytical equations Eq (1) compared with actual simulated weights at different time steps. The linear regression between the predicted values and the actual final values at the end of the stimulation is closer in burst firing compared to tonic firing ($R^2$ = 0.9983 in burst and $R^2$ = 0.9110 in tonic).

We test these predictions in the generic network shown in Fig 1A, consisting of 50 presynaptic and postsynaptic neurons. The network operates either in tonic mode, with random inputs (0.1–50 Hz), or in burst mode, induced by hyperpolarizing the inhibitory neuron. For details on the computational experiment, see Section Computational experiment related to Fig 2.

During tonic firing, inputs are uncorrelated, so each synapse experiences its own calcium pattern. As shown in Fig 2B-C, calcium traces differ widely across synapses. The resulting effective times in potentiation and depression are broadly distributed (Fig 2D), and weights converge to a wide range of fixed points (Fig 2F). In contrast, burst firing synchronizes activity, producing similar calcium traces and correlations across the network (Fig 2E). Consequently, all synapses converge toward a narrow set of fixed points—the burst-induced attractor (Fig 2H).

Analytical predictions closely match simulated weights (Fig 2G–I), with stronger correspondence in bursts ($R^2 = 0.9983$) compared to tonic activity ($R^2 = 0.9110$). The lower $R^2$ in tonic reflects both finite-time and stochastic effects. Simulations are run for 50 s per state to keep tonic and burst epochs comparable and within biologically and computationally plausible timescales; within this window, some synapses are analytically predicted to converge toward low values but do so only slowly, which increases residuals. In addition, tonic firing is driven by distinct noisy input spike trains, leading to heterogeneous calcium dynamics and variability in the effective time spent in potentiation and depression across synapses.

In contrast, collective burst firing synchronizes spike trains and calcium fluctuations across the network, producing more homogeneous plasticity drive and faster convergence toward the analytically predicted attractor. The STDP model shows the same pattern, lower $R^2$ in tonic ($R^2 = 0.3104$) and higher in burst ($R^2 = 0.9747$), for analogous reasons. Notably, the convergence demonstrated here characterizes the direction and structure of synaptic drift under burst firing; in shorter burst epochs, weights would be expected to move partially toward the same attractor, potentially interacting with slower consolidation mechanisms. In other neuromodulatory or biophysical regimes, transitions into burst firing could additionally rescale plasticity parameters (and thus the effective learning rate), so that synapses are still drawn toward the same burst-induced attractor but reach it on faster or slower timescales than those illustrated here.

Together, these results show that burst firing creates an attractor in weight space because shared activity patterns synchronize the effective plasticity drive across synapses, forcing them toward similar fixed points. Tonic firing, by contrast, disperses weights because each synapse follows its own independent activity statistics.

## Burst-induced attractor sensitivity to rhythmic state modulation

We next apply our analytical result to show how the burst-induced attractor can be modulated in practice, turning the apparent "reset" into a flexible mechanism for consolidation. Our framework predicts that altering the collective bursting dynamics modifies the effective time spent in potentiation and depression regimes (in calcium-based rules) or the spike-time correlations (in spike-based rules). As a result, the location of the attractor can be shifted without any other alteration.

To illustrate this, we return to the network described in Fig 1A. We emulate changes in the neuromodulatory drive by varying the external current applied to the inhibitory neuron across several values (Fig 3A, $I_{app,inh}$; the values used in each epoch are indicated above the pink trace and along the $x$-axis in Fig 3B). This manipulation mimics changes in neuromodulatory drive and produces distinct patterns of collective bursting activity, as seen in the raster plots and LFP profiles (Fig 3A). For details on the computational experiment, see Section Computational experiment related to Fig 3.

Consistent with the analytical prediction, the burst-induced attractor shifts slightly depending on the bursting regime, yet synaptic weights still converge within a narrow region of weight space (Fig 3A, bottom). The magnitude and direction of this shift reflect changes in the structure of collective bursting rather than the external current itself: regimes with stronger or longer bursts generate calcium transients that more frequently cross potentiation thresholds, biasing the attractor toward higher values. Because intrinsic neuronal heterogeneity and network interactions shape burst synchrony and spike content, the attractor position emerges from the resulting burst dynamics rather than from a simple linear dependence on the inhibitory drive.

Fig 3B summarizes this effect by plotting, for each burst epoch, the mean (horizontal segments) together with their standard deviation (vertical bars) as a function of $I_{app,inh}$. Tonic firing leaves weights broadly spread between the minimum and maximum values (Fig 3A), whereas collective bursting compresses them toward a narrow attractor whose position shifts smoothly with $I_{app,inh}$, showing that bursting reorganizes synaptic weights in a structured and controllable way.

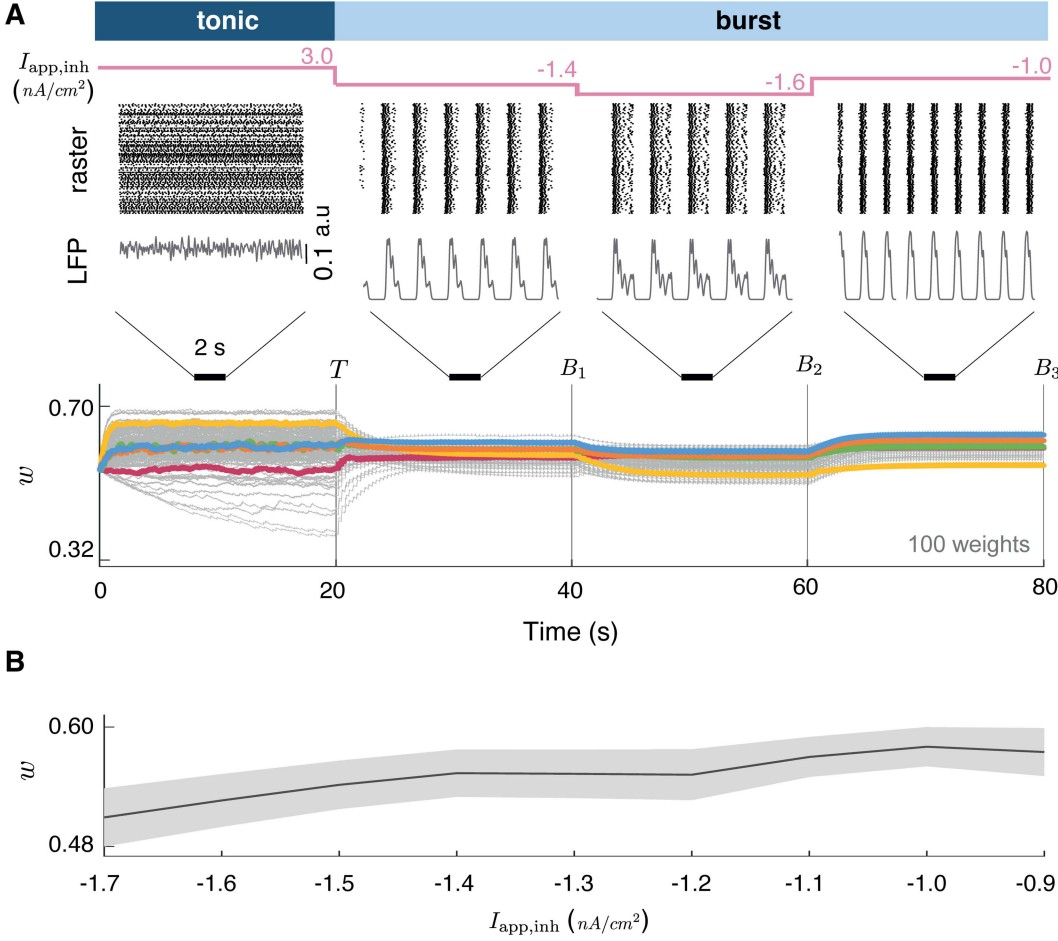

**Fig 3. Modulation of burst-induced attractor in synaptic weights. A.** Evolution of the network activity and synaptic weights when the network is driven from tonic to burst firing epochs with three variations of the collective burst firing (achieved by modulating the external current applied to the inhibitory neuron). T indicates the end of the tonic firing state, and $B_k$ indicates the end of the $k$-th burst firing state. 100 traces are randomly displayed among 2500 synaptic connections. The steady-state values associated can be modulated by the collective burst firing, moving the attractor in the weight space as a function of burst structure (e.g., spike content and duration). **B.** Summary of synaptic weights at the end of each burst epoch as a function of $I_{\mathrm{app,inh}}$. Vertical segments denote the standard deviation across all synapses, while horizontal bars indicate the mean weight. Tonic firing yields a broad spread of synaptic weights, whereas collective bursting compresses weights toward a narrow attractor whose location shifts smoothly with $I_{\mathrm{app,inh}}$.

## Neuromodulated and tag-dependent synaptic plasticity rules can split and shift the burst-induced attractor for memory consolidation

We can show the potential of the burst-induced attractor as a mechanism for synaptic consolidation and selection by applying our analytical result (Eq (1) and Eq (2)) to show how the burst-induced attractor can be modulated, in practice, by altering the plasticity parameters (e.g., $\Omega_p$ or $\Omega_d$ for calcium-based rule or $A^\pm$ or $\tau^\pm$ for STDP-like rule).

Recent evidence shows that synaptic plasticity is modulated by neuromodulators such as acetylcholine, dopamine, noradrenaline, serotonin, and histamine [31,41–51]. These neuromodulators influence plasticity at multiple stages of induction and consolidation [45,47,52,53]. Several computational models have incorporated neuromodulated plasticity rules [54,55]. For example, in sleep-dependent memory consolidation, the classical STDP kernel observed during wakefulness can be replaced with a purely depressive kernel during sleep, promoting synaptic down-selection [56]. Other models have shown that acetylcholine or dopamine can dynamically shift the STDP kernel to explain receptive field plasticity and

reward-driven navigation [57,58]. Additional frameworks introduce neuromodulation via eligibility traces or third-factor models [43,50,51].

Therefore, we model a simplified circuit of one inhibitory neuron connected to two excitatory neurons. Two consecutive 30-second periods of tonic and burst firing were simulated. Six circuit instances were tested: three with highly correlated excitatory firing (25–60 Hz) and three with low correlation ($\leq$ 10 Hz). For details on the computational experiment, see Section Computational experiment related to Fig 4. During the second burst epoch, the potentiation level $\Omega_p$ was reduced from 0.65 to 0.2, producing a global shift in the burst-induced attractor (Fig 4B). By modulating these parameters, neuro-modulators can shift the burst-induced attractor in the synaptic weight space. This effect can be replicated in spike-based rules by decreasing the amplitudes of potentiation ($A^+$) and depression ($A^-$) in the STDP kernel (S3 Fig). Dynamically, the same shifted attractors would be obtained if one started from an arbitrary heterogeneous weight distribution and inte-grated the burst-regime plasticity with these neuromodulated parameters; here, the preceding tonic phase simply provides a plausible way to generate such heterogeneity.

We next apply weight-dependent parameter changes only to synapses that exceed a tagging threshold at the end of the preceding tonic phase. Tagged synapse experience enhanced potentiation parameters ($\Omega_p$ = 95), while untagged syn-apses are downscaled ($\Omega_p$ = 0.2).

This selective modulation stabilizes tagged synapses near the higher values attractor and drives untagged synapses toward the lower one, leading to a bimodal distribution (Fig 4D). This tag-dependent burst-induced attractor implements selective stabilization for strong synapses while weakening others, a computational realization consistent with synaptic tagging and capture [59–62]. In principle, the same bimodal consolidation could be obtained by assigning tags directly to

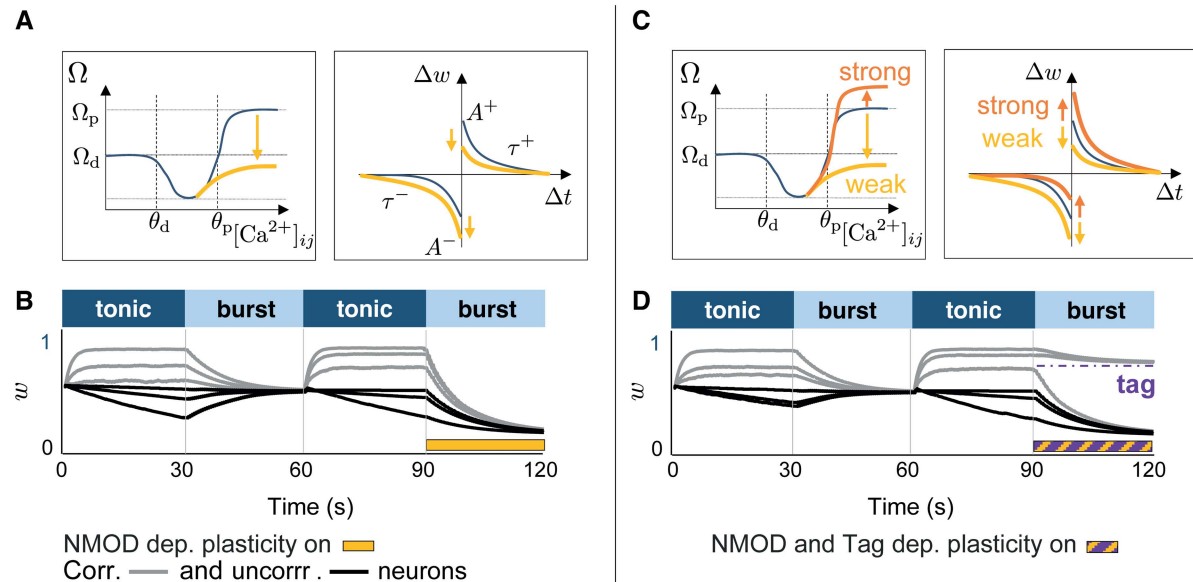

**Fig 4. Neuromodulated and tag-dependent synaptic plasticity rules exploit the burst-induced attractor for memory consolidation. A.** Calcium-based (left) and spike-based (right) plasticity rules, in which potentiation and depression parameters ($\Omega_p$, $A^+$, $A^-$) are modified by neuromodula-tion. **B.** Evolution of synaptic weights in six circuits subjected to two tonic–burst epochs. In the last burst epoch, the plasticity rule is down-neuromodulated (yellow bar), leading to a different lower burst-induced attractor. Half of the circuits display correlated excitatory activity (gray) and half uncorrelated activity (black). **C.** Calcium-based (left) and spike-based (right) plasticity rules with tag-dependent neuromodulation. Tagged synapses (orange) receive up-neuromodulation, while untagged synapses (yellow) receive down-neuromodulation. **D.** Same protocol as in **B**, but with tag-dependent neuromodulation during the last burst epoch (yellow–purple hatched bar). Synapses tagged during the tonic state (weights above threshold) converge to a higher burst-induced attractor, while untagged synapses converge to a lower one, resulting in bimodal weight consolidation.

an initial weight distribution and then applying burst-regime plasticity, but in our protocol the tonic phase provides a natural setting for tag formation.

Importantly, this process is not limited to bimodal outcomes. Because the attractor depends directly on the chosen parameters, any mapping from synaptic weight to parameter values can produce different target distributions during bursting. The resulting distribution can therefore be unimodal, bimodal, multimodal, or graded.

Together, these results show that neuromodulation and tagging make the apparent reset during bursting a flexible consolidation stage. Bursting compresses weights into an attractor, but the attractor itself can be shifted or split into several stable states depending on parameter changes.

## Discussion

In this work, we investigate how tonic and burst firing differentially shape synaptic plasticity within a biophysical network model capable of robustly switching between these two activity regimes [5]. We endow this network with a calcium-based plasticity rule [22,25] to explore how each regime contributes to the organization of synaptic weights. We also compare different synaptic plasticity rules, such as pair-based or triplet, to generalize our outcomes [19,20]. While tonic and burst firing are often studied in isolation, biological networks frequently alternate between them or express both within the same population, as seen in thalamic relay cells, hippocampal pyramidal neurons, and central pattern generators [1,7]. The interaction between these modes in shaping long-term synaptic organization remains largely unexplored, both experimentally and computationally, especially in networks that naturally alternate between tonic and burst firing. Our model provides a controlled framework to map how each regime influences the geometry of weight space.

Our modeling approach is intentionally simple but biologically grounded. The tonic and burst firing regimes considered here correspond to well-established modes observed across many neural circuits, from cortex and thalamus to hippocampus and brainstem, and are commonly linked to changes in intrinsic conductances and inhibitory control. Rather than modeling a specific experimental preparation, we aim to capture dynamical ingredients shared across systems to identify a mechanism by which firing regime specific plasticity dynamics organize synaptic weights.

Our results show that collective burst firing drives synaptic weights toward a narrow region in the weight space, a burst-induced attractor, whereas tonic firing maintains a more dispersed distribution. This convergence arises from synchrony-induced similarity in calcium dynamics or spike-time correlations across synapses, constraining their trajectories toward a shared attractor (Fig 5A). In contrast, tonic firing generates more heterogeneous activity patterns and multiple, separated convergence points. The mechanism is general, emerging across calcium-based and spike-based plasticity rules [18,20,22,23,25].

We further show that the location of the burst-induced attractor depends on both neural dynamics and the parameters controlling potentiation and depression. Neuromodulatory changes to these parameters can shift the attractor, enabling global reorganization during rhythmic bursting [47,52,55,57]. Implementing a tag-dependent rule, in which strong and weak synapses follow distinct neuromodulated kernels during bursting, creates two coexisting attractors: one stabilizing strong synapses and another downscaling weaker ones. This mechanism parallels the synaptic tagging and capture framework [59,60,63], offering a computational route for selective long-term consolidation (Fig 5B). More generally, because the attractor is determined by plasticity parameters, neuromodulation and tagging can generate not only two, but potentially multiple stable consolidation attractors, allowing different synaptic selection profiles.

An interesting extension is to incorporate additional hidden synaptic variables, such as eligibility traces or biochemical tags, which can store information independently of the expressed weight [50,64–66]. In such models, burst firing could pull the primary weight into the attractor while using hidden traces to transfer the acquired learning, allowing the coexistence of flexibility and long-term stability.

Overall, we identify burst-induced attractors as an emergent property of collective bursting combined with soft-bound plasticity rules. By demonstrating how neuromodulation and tagging can shift or split these attractors, we outline a

**A**

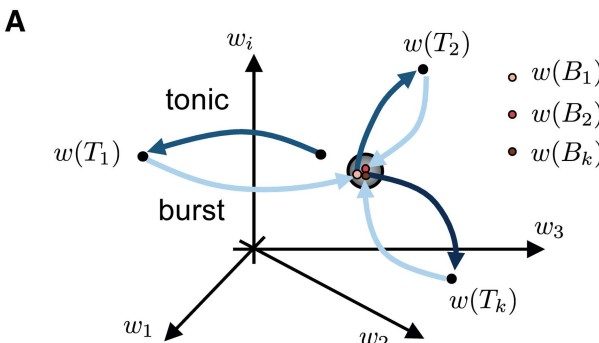

**B**

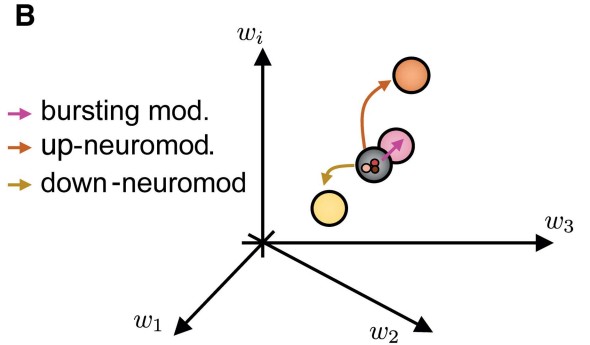

**Fig 5. Conceptual illustration of the burst-induced attractor and its modulation. A.** Example trajectories in synaptic weight space during transitions between tonic (light blue) and burst firing (dark blue). Tonic firing leads to more dispersed convergence points $w(T_k)$, whereas burst firing drives weights toward a narrower attractor region $w(B_k)$. **B.** Neuromodulation shifts the location of the burst-induced attractor; either by modulating the collective burst firing (pink, referring to Fig 3), or by modulating the plasticity parameters in a weight-dependent manner mimicking synaptic tag (referring to Fig 4). Up-neuromodulation moves the attractor toward higher weights (orange), while down-neuromodulation shifts it toward lower weights (yellow).

computational pathway for synaptic selection and consolidation. Tonic and burst firing each impose their own plasticity dynamics, while alternation between them determines the temporal sequence in which learning-like and consolidation-like processes are engaged. Extending this framework with hidden synaptic traces may help reconcile the trade-off between adapting to new inputs and preserving past memories—a balance that biological networks likely exploit [63]. This work further suggests that similar principles may extend to other firing regimes.

## Materials and methods

All original data in this work were generated programmatically using the Julia programming language [67]. Analyses were performed in Matlab. The code files are freely available at https://github.com/KJacquerie/Burst-Attractor. Any additional information required to reanalyze the data reported in this paper is available from the lead contact upon request.

## Computational models

**Neuron and network model.** All neurons are modeled using a single-compartment conductance-based using Hodgkin-Huxley formalism [68]. The membrane voltage $V$ of a neuron evolves according to the equation described by [5,30]:

$$C_m \dot{V} = -I_{leak} - \sum_{ion \in \mathcal{I}} I_{ion} - \sum_{p \in \mathcal{P}} \sum_{syn \in \mathcal{S}} I_{syn,p} + I_{app},$$

where $C_m$ represents the membrane capacitance, $I_{leak}$ is the leak current, $I_{ion}$ are the intrinsic ionic currents with $\mathcal{I}$ the set of all ionic channels, $I_{syn,p}$ are the synaptic currents with $\mathcal{S}$ the set of all synaptic neurotransmitter types and $\mathcal{P}$ the set of all presynaptic neurons, and $I_{app}$ denotes the applied current. Details about the ionic and synaptic currents and their associated dynamics and parameters are provided in S1 Text.

The neuronal network comprises an inhibitory neuron ($\mathcal{N}_{inh}$) projecting onto all excitatory neurons through $GABA_A$ and $GABA_B$ connections (refer to Fig 1B). Excitatory neurons are interconnected via a feedforward AMPA synapse, where the presynaptic neurons ($\mathcal{N}_{pre}$) influence the postsynaptic neurons ($\mathcal{N}_{post}$). The number of excitatory neurons varies across different computational experiments. The excitatory synaptic current perceived by the postsynaptic neuron $i$ from presynaptic neuron $j$ is characterized by:

$$I_{AMPA,ij} = w_{ij} \cdot \bar{g}_{AMPA,ij} \cdot s_{AMPA,j} \cdot (V_i - E_{AMPA}),$$

where $w_{ij}$ represents the synaptic weight, $\bar{g}_{AMPA,ij}$ represent the maximal conductance of the AMPA postsynaptic receptor (AMPAr). The variable $s_{AMPA,j}$ denotes the gating variable of the AMPAr, dynamically modulated by the presynaptic membrane voltage ($V_j$) and $E_{AMPA}$ is the reversal potential of AMPAr.

### Switch from tonic to burst firing

The inhibitory neuron dynamically influences the network activity. An external current applied to the inhibitory neuron ($I_{app,inh}$) is used to control the firing regime of the network, mimicking the effect of neuromodulatory signals. A depolarizing current induces tonic firing activity in this inhibitory neuron, causing all excitatory cells to remain at their resting potential until stimulated. A sufficiently large external depolarizing pulse can evoke action potentials in excitatory neurons. Conversely, a hyperpolarizing current applied to the inhibitory neuron switches the entire network into a synchronized collective bursting activity throughout the network [5,69]. In each computational experiment, the depolarizing current $I_{app,inh}$ is equal to 3 nA/cm$^2$ for tonic state and -1.2 nA/cm$^2$ for burst state.

The use of a single inhibitory neuron to control network state is not intended as a literal microcircuit model, but as a coarse-grained representation of state-dependent inhibitory or neuromodulatory control of network activity. Similar low-dimensional control motifs are commonly used in models of rhythmic bursting and brain-state transitions, where inhibition determines whether excitatory populations operate in tonic or bursting regimes [70–72]. In our setting, hyperpolarizing this inhibitory neuron unmasks intrinsic bursting dynamics in the excitatory population, rather than imposing bursts through externally patterned input.

### External applied pulse train current

In tonic firing states, each excitatory neuron $i$ is triggered by an applied current ($I_{app,i}$) to make the neuron fire at a nominal frequency $f_0$. To generate this input-driven tonic activity, a neuron receives a pulse train current, where each pulse lasts 3 ms and has an amplitude that is independently sampled from a uniform distribution on an interval between 50 nA/cm$^2$ and 60 nA/cm$^2$. The interpulse intervals (*i.e.*, the time between two successive pulses) are independently sampled from a Normal distribution with a mean equal to $1/f_0$ and a standard deviation equal to $0.1/f_0$.

In burst firing states, each excitatory neuron $i$ undergoes a noisy baseline such as the applied current ($I_{app,i}$) randomly fluctuating between the interval 0 and 1 nA/cm$^2$. This amplitude is chosen to remain well below the levels required to elicit spikes during tonic firing, so it acts as background noise rather than a driver of bursting; the rhythmic burst pattern therefore arises from intrinsic conductances and network interactions, not from an externally imposed oscillatory input.

### Homogeneous and heterogeneous network

We define a *homogeneous* network (or circuit), where each neuron has the same intrinsic ion channel maximal conductances $\bar{g}_{ion} = \bar{g}_{ion}^*$ (numerical values provided in S1 Text). We define a *heterogeneous* network (or circuit), where the

maximal conductance $\bar{g}_{ion}$ for each neuron is randomly chosen within a $\pm 10\%$ interval around its nominal value $\bar{g}_{ion}^*$, such as $\bar{g}_{ion} = \bar{g}_{ion}^* \cdot (1 + \epsilon)$ with $\epsilon \sim \text{Unif}(-0.1, 0.1)$.

## Local field potential

The local field potential (LFP) measures the average behavior of interacting neurons. It reflects the collective excitatory synaptic activity received by the postsynaptic neuron population. The overall synaptic activity is measured by the mean of the individual synaptic currents:

$$\text{LFP}(t) = -\frac{1}{M}\sum_{j=1}^{M}\sum_{i=1}^{N} I_{\text{AMPA},ij}(t),$$

where $M$ is the number of postsynaptic neurons and $N$ is the number of presynaptic neurons [5,30].

## Synaptic plasticity

**Calcium-based model.** The change in the synaptic weight $w_{ij}$ between a presynaptic neuron $j$ and a postsynaptic neuron $i$ is governed by the calcium-based model proposed by [25]:

$$\tau_w \dot{w}_{ij} = -w_{ij}(1 - w_{ij})(w^* - w_{ij}) - \gamma_d w_{ij}\Theta(c_{ij} - \theta_d) + \gamma_p(1 - w_{ij})\Theta(c_{ij} - \theta_p). \tag{3}$$

Here, $\tau_w$ represents the time constant, $w^*$ defines the stable state (equal to 0.5), $\gamma_p$ is the potentiation rate, $\gamma_d$ is the depression rate, $\theta_p$ is the potentiation threshold, and $\theta_d$ is the depression threshold. The function $\Theta(\cdot)$ is the Heaviside function, which returns 1 when the argument is positive and 0 otherwise.

The change in synaptic weight depends on the calcium concentration $c_{ij}$, which is the sum of the calcium caused by the activity of the presynaptic neuron $j$ and the activity of the postsynaptic neuron $i$. A pre- or postsynaptic spike translates into a calcium exponential decay. Further explanations are provided in S1-B Text. The calcium-based rule defined by [25] implements a soft-bound, where the perceived potentiation rate $\gamma_p(1 - w_{ij})$ is smaller for high $w_{ij}$ than for low $w_{ij}$. The same reasoning applies to the depression rate $\gamma_d$. Detailed parameter values are provided in S1 Text C.

In Fig 2–4, the calcium-based model is modified according to the version of [22]. The model is similar except that the stable state is removed and the fitted plasticity parameters are adapted:

$$\tau_w \dot{w}_{ij} = \gamma_p(1 - w_{ij})\Theta(c_{ij} - \theta_p) - \gamma_d w_{ij}\Theta(c_{ij} - \theta_d). \tag{4}$$

The only difference between the two calcium-based models is that when calcium levels fall below the depression threshold, the synaptic weight in the 2016 model is no longer subject to the cubic term and therefore remains unchanged. Both rules can be applied, and in either case the burst-induced attractor is present. However, the 2016 formulation is more convenient for demonstrating this phenomenon.

## Computational experiment related to Fig 1

Fig 1A-B illustrate the activity in the feedforward network, comprising 50 presynaptic neurons connected to 50 postsynaptic neurons. The network is heterogeneous. The network is in tonic firing mode during 1.5 s. Each excitatory neuron receives a train of current pulses with nominal frequencies $f_0$ randomly sampled from a uniform distribution between 0.1 and 50 Hz. Then, the network switches to synchronized collective bursting during 2.5 s.

Fig 1C illustrates the evolution of the weights, consisting of 50 presynaptic neurons connected to 50 postsynaptic neurons, along with one inhibitory neuron projecting GABA current onto all of them. It comprises a total of 2500 weights.

This computational experiment consists of 8 states, interleaving tonic and burst firing, each lasting 20 s. During tonic firing, neurons are stimulated with a pulse train current, where the pulse frequency for each neuron is randomly chosen from a uniform distribution between 0.1 Hz and 50 Hz.

Fig 1D demonstrates the values of the normalized weights at the end of the fourth tonic state (vertical axis) with the values of the normalized weights at the end of the third tonic state (horizontal axis). The weights are normalized using the formula

$$\tilde{w}_{ij}(t) = \frac{w_{ij}(t) - \min_{i,j,t}(w_{ij}(t))}{\max_{i,j,t}(w_{ij}(t)) - \min_{i,j,t}(w_{ij}(t))},$$

where $\max_{i,j,t}(w_{ij}(t))$ and $\min_{i,j,t}(w_{ij}(t))$ denote the maximum and minimum values obtained by comparing the values at the end of the third and fourth tonic states. The distribution of the normalized weights is represented for bins of width 0.025 between 0 and 1. Similarly, for burst firing (Fig 1E), the same data analysis protocol is used for the synaptic weights at the end of the third burst state and the end of the fourth burst state. The scatter plots computed for the other states are shown in S1 Fig.

### Computational experiment related to Fig 2

Fig 2A-C illustrate the calcium fluctuations between two presynaptic neuron *j* connected to one postsynaptic neuron *i*. The calcium dynamics are following [25] model, (iii) [22,25] (see S1 Text). Fig 2D-E are the time spent in each calcium region (see The burst-induced attractor emerges in a wide range of synaptic plasticity models for calculation). Fig 2F-H illustrate the evolution of the synaptic weights during two states of 50 s within the same network of 50 pre- and 50 postsynaptic neurons for 100 weights among 2500. The duration of 50 s per state was chosen as a compromise between biological plausibility and computational cost, and to allow a direct comparison between tonic and burst regimes under matched time windows. The network has the same set of intrinsic and synaptic conductances as in Fig 1. The two states are initialized with the same set of initial conductances chosen randomly within 0 and 1. Fig 2G-I are the comparison of the predicted value given by the Eq (1) and the simulated value directly extracted during the simulation.

### Computational experiment related to Fig 3

Fig 3A (bottom) illustrates the evolution of the synaptic weights during four states of 20 s within the same network of 50 presynaptic and 50 postsynaptic neurons, together with one inhibitory neuron, maintaining identical maximal conductances as in Fig 1. The first state corresponds to a tonic state, where the inhibitory neuron is depolarized at 3 nA/cm² during 20 s. The excitatory neurons receive a pulse train current with the pulse frequency randomly chosen from a uniform distribution between 0.1 Hz and 50 Hz for each neuron. This tonic state is succeeded by three bursting states, each lasting 20 s. The hyperpolarizing current was varied across multiple values in a hyperpolarized range (from approximately -1.7 nA/cm² to -0.9 nA/cm²), as indicated in Fig 3A and 3B, thereby producing distinct collective bursting regimes.

Fig 3B displays the distributions of the synaptic weights at the end of each burst state. The synaptic weights are not normalized, and the distribution is computed for bins of width 0.025 between 0 and 1.

### Computational experiment related to Fig 4

Fig 4B shows the evolution of synaptic weights over four 30 s states in six homogeneous circuits. Each circuit consists of one inhibitory neuron connected to two excitatory neurons. During the tonic firing states, three circuits are configured with correlated excitatory activity, where the pulse frequency is selected between 30 Hz and 60 Hz, and three circuits are configured with uncorrelated activity, with pulse frequencies between 0 Hz and 10 Hz. This setup allows us to sample a broad range of initial weight trajectories, from potentiation to depression. In the second burst state, the potentiation level $\Omega_p$ is reduced from 0.65 to 0.2.

Fig 4D uses the same circuit configuration and parameters, except that at the end of the second tonic state, synapses are tagged according to their weight values. Synapses with $w > 0.75$ at the end of the tonic state are assigned an up-neuromodulated calcium-based rule with $\Omega_p = 0.95$, while the remaining synapses follow a down-neuromodulated rule with $\Omega_p = 0.2$.

## Supporting information

**S1 Text. A. Conductance-based model description; description of the neuron model and the network architecture.** B. Synaptic plasticity implementation. (i) Description of the calcium-based models starting with the calcium dynamics, (ii) [25] model, (iii) [22] model, (iv) [21] model, (v) [23] model, (iv) implementation of hard bounds, (v) description of the spike-time dependent models starting with the pair-based model, (vi) Triplet model. C. Computational experiments: numerical values. D. Derivation of the burst-induced attractor in a calcium- based model. E. Derivation of the burst-induced attractor in a spike-time dependent plasticity rule. F. Derivation of the burst-induced attractor in a model using hard-bounds.
(PDF)

**S1 Fig. Replication of the analysis illustrated in Fig 1D-E at different tonic and burst states.**
(PDF)

**S2 Fig. Replication of the experiment illustrated in Fig 1C in various synaptic plasticity rules using soft-bounds (left panel) and hard-bounds (right panel).** Comparison of the synaptic weights at the end of the third and fourth tonic firing states (left column) or the third and fourth burst firing states (right column), normalized between the minimal and maximal values. (CTX = cortex, data fitted on [35]; HPC = hippocampus, data fitted on [38]).
(PDF)

**S3 Fig. Additional results associated with Fig 4.** A. Global neuromodulation. Left: plasticity rules in spike-based (top) and calcium-based (bottom) formulations, where potentiation and depression parameters ($A^p$, $A^m$, or $\Omega_p$, $\Omega_d$) are globally downscaled (yellow arrows). Right: synaptic weights $w$ converge toward a lower attractor during bursting, showing that global parameter changes shift the attractor in weight space. B. Tag-dependent modulation. Left: plasticity rules where synapses above a tagging threshold receive enhanced potentiation parameters, while weaker synapses are downscaled. Right: tagged synapses converge to a higher attractor (purple dashed line), while untagged synapses decay toward a lower one, resulting in bimodal consolidation. This mechanism shows how neuromodulation and tagging can selectively stabilize strong synapses while weakening weaker ones. This figure shows the evolution of synaptic weights between two excitatory neurons during burst firing under different initial conditions (0:0.1:1). In blue, trajectories correspond to unmodulated plasticity parameters, while in yellow they show neuromodulated parameters. The color gradient emphasizes the initial strength of the synaptic weights, with darker shades indicating larger initial values.
(PDF)

## Acknowledgments

The authors thank Nora Benghalem, Juliette Ponnet, and Caroline Minne for their help in the early stages of the project. The authors acknowledge the valuable insights and feedback from Professor Eve Marder on the project.

## Author contributions

**Conceptualization:** Kathleen Jacquerie, Guillaume Drion.

**Funding acquisition:** Kathleen Jacquerie, Pierre Sacré, Guillaume Drion.

**Methodology:** Kathleen Jacquerie, Danil Tyulmankov, Guillaume Drion.

**Software:** Kathleen Jacquerie.

**Supervision:** Danil Tyulmankov, Pierre Sacré, Guillaume Drion.

**Visualization:** Kathleen Jacquerie.

**Writing – original draft:** Kathleen Jacquerie.

**Writing – review & editing:** Kathleen Jacquerie, Danil Tyulmankov, Pierre Sacré, Guillaume Drion.

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
