## [Decision Letter · Decision Letter 0]

29 Oct 2025

Switching from tonic to burst firing creates an attractor in synaptic weight dynamics

PLOS Computational Biology

Dear Dr. Jacquerie,

Thank you for submitting your manuscript to PLOS Computational Biology. After careful consideration, we feel that it has merit but does not fully meet PLOS Computational Biology's publication criteria as it currently stands. Therefore, we invite you to submit a revised version of the manuscript that addresses the points raised during the review process.

Please submit your revised manuscript within 60 days Dec 28 2025 11:59PM. If you will need more time than this to complete your revisions, please reply to this message or contact the journal office at ploscompbiol@plos.org. Please include the following items when submitting your revised manuscript:

We look forward to receiving your revised manuscript.

Kind regards,

Jian Liu

Academic Editor

PLOS Computational Biology

Thomas Serre

Section Editor

PLOS Computational Biology

**Additional Editor Comments:**

This work presents a study on tonic and burst firing neuronal dynamics. To strengthen the manuscript, the authors should provide a more detailed justification for the model's assumptions and a more rigorous validation of its dynamics against experimental data, at least by providing more discussions.

**Journal Requirements:**

3) We notice that your supplementary Figures are included in the manuscript file. Please remove them and upload them with the file type 'Supporting Information'. Please ensure that each Supporting Information file has a legend listed in the manuscript after the references list.

4) Please amend your detailed Financial Disclosure statement. This is published with the article. It must therefore be completed in full sentences and contain the exact wording you wish to be published.

1) Please clarify all sources of financial support for your study. List the grants, grant numbers, and organizations that funded your study, including funding received from your institution. Please note that suppliers of material support, including research materials, should be recognized in the Acknowledgements section rather than in the Financial Disclosure

2) State the initials, alongside each funding source, of each author to receive each grant. For example: "This work was supported by the National Institutes of Health (####### to AM; ###### to CJ) and the National Science Foundation (###### to AM)."

3) State what role the funders took in the study. If the funders had no role in your study, please state: "The funders had no role in study design, data collection and analysis, decision to publish, or preparation of the manuscript."

4) If any authors received a salary from any of your funders, please state which authors and which funders..

**Reviewers' comments:**

Reviewer's Responses to Questions

**Comments to the Authors:**

Reviewer #1: Summary:

This paper studies the synaptic weight dynamics of a biophysically-inspired neural network model with synaptic plasticity during alternating firing regimes, namely tonic (background/asynchronous/steady state) and burst firing (rapid clusters/synchronous). It highlights that previous work has examined these two firing regimes separately, but that the current work brings novelty in its consideration of the consecutive switching between regimes, arguing that the regimes produce different synaptic weight dynamics, and could serve functionally important and complementary purposes. The authors use a combination of analysis and simulation to demonstrate that the tonic regime induces network flexibility via the stochasticity of firing and the resulting broad distribution of synaptic weights, while the burst phase induces network stability via the homogenisation/convergence of synaptic weights in a narrower distribution of guaranteed "attractor" states. They relate this to mechanisms for learning and memory, such as active sensory processing (during tonic firing) and memory consolidation (during burst firing). The paper shows via analysis and simulation that the fixed points of synaptic weights during the burst firing regime are dependent on the activity statistics and plasticity parameters, and it demonstrates that the results hold across various plasticity rules. The authors suggest that the dependence of the attractor on plasticity parameters and neural activity could be a learning and memory mechanism employed by synaptic tagging and capture, or neuromodulation. The paper is clear and well written, the model and methods are well-described, and the figures are explanatory and very clear. Code is made available and signposted.

Major:

The major question I would raise is whether this work in fact addresses the dynamics arising from SWITCHING between firing regimes, as claimed. The paper states that plasticity has previously “been studied only in either one of these regimes,” and that this study seeks to clarify how ALTERNATING between tonic and burst firing shapes long-term synaptic weight dynamics. However, it is not entirely clear that the ALTERNATION itself contributes new dynamical structure beyond what is already observed in each regime separately.

The model convincingly shows that tonic firing produces heterogeneous, broad weight distributions, and burst firing produces homogeneous, convergent weight states. Yet when these regimes are concatenated in alternation, the resulting dynamics appear to be merely the superposition of two independent phases: the weight evolution during each phase depends only on the instantaneous plasticity rule and activity statistics, not on the prior regime. Consequently, in the baseline model, the tonic phase has no causal influence on the subsequent burst phase (and vice versa). Only when additional mechanisms such as synaptic tagging or parameter modulation are introduced does the alternation acquire genuine dynamical significance. However, it could be argued that even these effects do not strictly require an explicit tonic phase: one could equivalently initialise the system with a (heterogeneous) distribution of weights, assign pre-tagged synapses/neuromodulation, and then evolve it under the burst-regime dynamics to obtain the same attractor differentiation. In that sense, the alternation between regimes is not essential for demonstrating the key mechanism; rather, it provides a plausible biological context for how such conditions might arise. I acknowledge that this contextual role is important, as the suggested link with possible learning and memory mechanisms (such as sensory processing, memory consolidation, and a "reset") rests conceptually on the alternation between regimes. Pointing out the distinction between the model’s dynamical necessity and its biological interpretation would clarify the paper’s central claim.

I would maybe suggest a slight reframing to emphasise that the alternation between regimes is what enables the proposed functional interpretations - linking distinct dynamical regimes to phases of learning and memory - rather than primary driver of distinct weight dynamics. In this view, the alternation provides the biological and computational context that makes these mechanisms relevant, even if the underlying plasticity dynamics arise locally within each regime.

Minor:

Just something I want to clarify: on lines 157 to 159 it is stated that the bursting activity "is independent of external input currents $I_{\text{app},j}$ and $I_{\text{app},i}$", but in Fig1B, $I_{\text{app},j}$ is off/silent. It is not fully clear whether bursting would remain independent if $I_{\text{app},j}$ continued to deliver pulses, as in the tonic phase. Methods describe $I_{\text{app},i}$ as fluctuating between 0 and 1 nA/cm2, but this does not say anything about the independence of the bursting activity from input. Currently, the figure demonstrates independence from SILENCE, but not from ACTIVE INPUT. It is only a minor clarification, and would not impact the results or conclusions.

In section 4.5 it is mentioned that the $R^2$ in tonic firing is lower than in burst firing due to "finite-time effects" (i.e. "some synapses are predicted to depress toward zero but have not yet completed that trajectory within the 50 s window, leading to larger residuals"). This raises a practical question: Why not extend the window to reduce the finite-time effects? How long would you need to simulate tonic firing to achieve comparable convergence to burst firing? Also, the authors could consider the fact that maybe the noise level/stochasticity of the tonic phase also contributes (a lot?) to the lower $R^2$; this might be worth adding.

Relatedly, the timescales over which the attractor forms could be important for your interpretations/conclusions. Understanding this relationship (e.g. how rates of convergence depend not only on plasticity parameters but also on the initial weight distribution) could provide insight into the functional relevance of bursts, such as how long bursting activity is required to stabilise synapses or how tonic-induced variability shapes consolidation dynamics. Perhaps the authors can comment on whether heterogeneous timescales in attractor formation are present in the model and if this has a bearing on the functional interpretations?

In section 4.6 the authors point out that modulation of external drive can result in the attractor shifting "slightly", depending on the bursting regime. Is there a way of quantifying or making the degree of "shift" more precise? The 3 different attractors in Fig 3A are arguably noticeably different. Being able to quantify the shift would allow the reader to understand how slight or substantial it is, and permit one to measure how sensitive the attractor is to the external current, to understand if the relationship is linear or not.

Typos:

line 144 spelling "goverining" to "governing"

line 307 change "untagges" to "untagged"

line 309 "yieleding" you mean "yielding", or "leading to"?

309-310 "tag-depdent" to "tag-dependent"

line 310 "weaking" change to "weakening"

line 351 "profile" to "profiles"?

last sentence of page 18 "The smallest, the slowest the kinetics" - unsure what that is meant to be. The sentence needs a verb, at least. E.g. "The smaller the parameters are, the slower the kinetics will be".

line 453, reference is made to Fig 4E, which does not exist; most likely Fig 1E is meant.

References:

Bi Gq, Poo Mm (2001) => Bi GQ, Poo MM (consistent with your other references to their work)

Reviewer #2: The study is motivated by the question that how synaptic weights evolve when activity shifts between tonic and burst firing. The study shifts the tonic and burst firing of excitatory neurons by modulating the current injected into the inhibitory neuron shared by those excitatory neurons. They found the burst firing tends to synchronize the learned E weights, i.e., the E weights will converge to similar values, while the E weights learned during the tonic mode will be much more diverse. They also analytically derived the eventual weights are weighted average of the stationary synaptic E weights learned under both tonic and burst weights respectively, with the mixing weight the proportion of time spent in each firing mode.

Major:

An important conceptual question is the time scale of synaptic plasticity is far longer than the switching between tonic and burst. The figures indicate the weights only take few dozens of seconds to reach stationary, which should not be the case in reality. Although I understand this is probably a math simplification to speed up the simulation and doesn’t affect the conclusion substantially, it will be important for the author to explicitly explain this assumption.

In reality, the time spent of burst firing mode is short, unlike the sustained bursting firing lasting for 20s in the simulation. My understanding of the sustain burst firing in the simulation is only for math analysis. Nevertheless, considering the few moments of burst firing observed in reality, I wonder how much the burst firing play a significant role in shaping learned synaptic weights? It worths some discussions.

Technical:

I haven’t checked the math derivations line by line but based on the comparison of math predictions and simulations (e.g., Fig. 2G & I, etc.) I believe the correctness of the theoretical analysis (Eqs. 1-2) and they are consistent with intuition.

Clarity:

The manuscript is organized well, and the figures are clear.

**Have the authors made all data and (if applicable) computational code underlying the findings in their manuscript fully available?**

The PLOS Data policy requires authors to make all data and code underlying the findings described in their manuscript fully available without restriction, with rare exception (please refer to the Data Availability Statement in the manuscript PDF file). The data and code should be provided as part of the manuscript or its supporting information, or deposited to a public repository. For example, in addition to summary statistics, the data points behind means, medians and variance measures should be available. If there are restrictions on publicly sharing data or code —e.g. participant privacy or use of data from a third party—those must be specified.requires authors to make all data and code underlying the findings described in their manuscript fully available without restriction, with rare exception (please refer to the Data Availability Statement in the manuscript PDF file). The data and code should be provided as part of the manuscript or its supporting information, or deposited to a public repository. For example, in addition to summary statistics, the data points behind means, medians and variance measures should be available. If there are restrictions on publicly sharing data or code —e.g. participant privacy or use of data from a third party—those must be specified.

Reviewer #1: Yes

Reviewer #2: Yes

PLOS authors have the option to publish the peer review history of their article (what does this mean? ). If published, this will include your full peer review and any attached files.). If published, this will include your full peer review and any attached files.

**Do you want your identity to be public for this peer review?** For information about this choice, including consent withdrawal, please see our For information about this choice, including consent withdrawal, please see our Privacy Policy ..

Reviewer #1: No

Reviewer #2: **Yes:** Wenhao ZhangWenhao Zhang

**Figure resubmission:**
---

## [Decision Letter · Decision Letter 1]

9 Feb 2026

Dear Jacquerie,

We are pleased to inform you that your manuscript 'Burst firing creates an attractor in synaptic weight dynamics' has been provisionally accepted for publication in PLOS Computational Biology.

Best regards,

Jian Liu

Academic Editor

PLOS Computational Biology

Thomas Serre

Section Editor

PLOS Computational Biology

Reviewer's Responses to Questions

**Comments to the Authors:**

Reviewer #1: The authors have addressed all my points clearly and satisfactorily. I have no additional comments and recommend acceptance. This paper will be of interest to a broad audience in computational neuroscience and computational biology. One small note: in the resubmission document, some in-text citations do not display correctly — this appears to be a formatting issue.

Reviewer #2: Thanks for the authors' reply and the revision of the manuscript. The reply and revision both address my concern and now I think the manuscript is ready to publish.

**Have the authors made all data and (if applicable) computational code underlying the findings in their manuscript fully available?**

The PLOS Data policy requires authors to make all data and code underlying the findings described in their manuscript fully available without restriction, with rare exception (please refer to the Data Availability Statement in the manuscript PDF file). The data and code should be provided as part of the manuscript or its supporting information, or deposited to a public repository. For example, in addition to summary statistics, the data points behind means, medians and variance measures should be available. If there are restrictions on publicly sharing data or code —e.g. participant privacy or use of data from a third party—those must be specified.requires authors to make all data and code underlying the findings described in their manuscript fully available without restriction, with rare exception (please refer to the Data Availability Statement in the manuscript PDF file). The data and code should be provided as part of the manuscript or its supporting information, or deposited to a public repository. For example, in addition to summary statistics, the data points behind means, medians and variance measures should be available. If there are restrictions on publicly sharing data or code —e.g. participant privacy or use of data from a third party—those must be specified.

Reviewer #1: Yes

Reviewer #2: Yes

PLOS authors have the option to publish the peer review history of their article (what does this mean? ). If published, this will include your full peer review and any attached files.). If published, this will include your full peer review and any attached files.

**Do you want your identity to be public for this peer review?** For information about this choice, including consent withdrawal, please see our For information about this choice, including consent withdrawal, please see our Privacy Policy ..

Reviewer #1: No

Reviewer #2: **Yes:** Wenhao ZhangWenhao Zhang

---

## [Editor Report · Acceptance letter]

PCOMPBIOL-D-25-01933R1

Burst firing creates an attractor in synaptic weight dynamics

Dear Dr Jacquerie,

I am pleased to inform you that your manuscript has been formally accepted for publication in PLOS Computational Biology. Your manuscript is now with our production department and you will be notified of the publication date in due course.

With kind regards,

Judit Kozma
